# A Longitudinal Observational Study to Monitor the Outpatient–Caregiver Dyad in a Rehabilitation Hospital: Sociodemographic Characteristics and the Impact of Cognitive and Functional Impairment

**DOI:** 10.3390/brainsci15121316

**Published:** 2025-12-10

**Authors:** Daniela Mancini, Valeria Torlaschi, Marina Maffoni, Roberto Maestri, Pierluigi Chimento, Michelangelo Buonocore, Antonia Pierobon, Cira Fundarò

**Affiliations:** 1Psychology Unit, Montescano Institute, Istituti Clinici Scientifici Maugeri IRCCS, 27040 Montescano, Italy; daniela.mancini@icsmaugeri.it (D.M.); valeria.torlaschi@icsmaugeri.it (V.T.); antonia.pierobon@icsmaugeri.it (A.P.); 2Department of Biomedical Engineering, Montescano Institute, Istituti Clinici Scientifici Maugeri IRCCS, 27040 Montescano, Italy; roberto.maestri@icsmaugeri.it; 3Neurophysiopathology Unit, Montescano Institute, Istituti Clinici Scientifici Maugeri IRCCS, 27040 Montescano, Italy; pierluigi.chimento@icsmaugeri.it (P.C.); michelangelo.buonocore@icsmaugeri.it (M.B.); cira.fundaro@icsmaugeri.it (C.F.)

**Keywords:** dyad, dementia, MCI, burden, caregiver, rehabilitation

## Abstract

**Background and objectives:** This study examines how sociodemographic, clinical, and psychological factors within the patient–caregiver dyad affect caregiver burden and health-related quality of life (HRQoL) in cognitive impairment. By comparing baseline data with a 1-year follow-up, the research aims to identify key predictors of caregiver burden and well-being. **Methods**: A longitudinal observational study was conducted in an Italian rehabilitation hospital, recruiting 132 outpatients and their caregivers at baseline, categorized as (a) Mild Cognitive Impairment (MCI, n = 33); (b) dementia (DEM, n = 58); (c) healthy subjects (No-CI, n = 41). One year after baseline assessment (T0), patients were contacted and invited for an in-person follow-up re-evaluation (T1). Most attrition was related to the COVID-19 pandemic. Statistical analyses included non-parametric tests for group comparisons and stepwise multiple linear regression to identify predictors of burden, adjusting for confounders (e.g., age, gender, education, employment, co-residence). **Results:** A total of 51 subjects (age: 80.0 ± 6.1) and 34 caregivers (age: 58.8 ± 15.9) were evaluated. Patients were balanced by gender (53% males); most were retired (96%), married (62.7%), and cared for by sons (47%) or wife–husband (47%). Caregivers (females: 85%) were married (68.3%) and active workers (46.4%). Over one year, 17 No-CI subjects developed MCI or DEM; 15 MCI patients progressed to DEM. Caregiver HRQoL negatively correlated with distress and burden in MCI and DEM groups. Patient cognitive status, functional abilities, neuropsychiatric symptoms, and gender predicted caregiver burden, emphasizing the interplay between clinical and demographic factors. **Conclusions:** It is essential to monitor psychosocial factors in both the patient and the caregiver to develop effective prevention and support strategies.

## 1. Introduction

Cognitive decline is a process characterized by a decrease in cognitive functions, which may be progressive in certain conditions (e.g., neurodegenerative diseases) or reversible and temporary in others (e.g., cognitive impairment related to major depressive disorder). Progressive cognitive decline encompasses a spectrum from mild forms, known as Mild Neurocognitive Disorder or Mild Cognitive Impairment (MCI), to severe forms of dementia named Major Neurocognitive Disorder [1]. Studies show that MCI may be a predictive factor for dementia, with 10% to 15% of cases developing the disease [2,3]. According to cognitive impairment severity, patients may also exhibit behavioral symptoms such as communication difficulties, personality changes, oppositionality, resistance to treatment, and decline in basic and instrumental daily activities such as bathing, cooking, eating, dressing, and shopping [4].

These cognitive and behavioral manifestations, particularly in MCI diagnosis and even more in dementia, heavily affect both patients and their families, with a noticeable impact on the primary caregiver in terms of emotion and care [5]. This phenomenon is named caregiver “burden”, referring to the level of caregivers’ impairment in their emotional or physical health, social interactions, and financial status due to the patient’s assistance [6].

Informal caregivers are increasingly required to assume growing responsibilities that intensify as the patient’s condition worsens [7], given the irreversible and degenerative nature of dementia. The literature shows that primary caregivers of people with dementia experience significantly higher levels of stress, anxiety, and depression when compared to those who do not care for people with cognitive impairment or those with other non-neurological chronic conditions [8]. The main informal caregiver may be the elderly partner, potentially frail and in poor health, or the middle-aged son/daughter caring for the elderly parents and young children at the same time [7,8]. Thus, the caregiver–patient relationship in the context of cognitive impairment is a relevant topic in the scientific literature and the healthcare landscape, particularly because of the significant demand for care from this population.

Current models of care emphasize active engagement from both patients and informal caregivers, highlighting two key aspects: the shared care dyad and a collaborative partnership between the caregiver and the individual with cognitive impairment, who is regarded as much as possible as an active partner in their own care [9]. Specifically, we can define it as a “family-centered care approach” as it significantly reduces caregiver burden for individuals with cognitive impairment through targeted interventions that address psychological and practical challenges of the dyad [9]. So, the taking-care process is collaborative, with shared responsibilities between the patient and her/his caregiver [10]. Doing so, the disease condition becomes a shared family experience [11]. Moreover, the caregivers’ perception of cognitive impairment significantly contributes to caregiver burden, so that a dyad-centered approach is necessary for support [9,10,11,12]. Despite the variety and complexity of dyadic interventions for older adults with chronic diseases, some approaches, like family-engaged Dignity Therapy, showed positive effects on hope, mental health, and family cohesion, while also reducing caregiver anxiety and depression [12].

According to this literature knowledge, our first study investigated the impact of cognitive impairment on caregiver–patient dyads, revealing a significant effect of cognitive disease on caregiver burden and quality of life [13]. Specifically, we found that caregivers’ burden and quality of life are influenced by patients’ autonomy level, reduced neuropsychiatric symptoms, and improved patient functioning [13].

Thus, to further explore the impact of different cognitive impairments on dyads over time, our second study aimed to describe socio-demographic, clinical, psychological, and health-related quality of life (HRQoL) variables between two groups of patients with MCI or dementia and their respective caregivers at baseline and at 1-year follow-up. We compare baseline with 1-year follow-up data, highlighting any changes in the sample and exploring variations in burden and HRQoL. Moreover, we assess the predictive value of some socio-demographic factors, cognitive conditions, daily living activities, and behavioral/emotional symptoms on the caregiver burden.

Our primary outcome is caregiver burden measured by the FSQ-SF score at one year. We hypothesize that higher burden at follow-up will be significantly predicted by patients’ baseline cognitive and functional decline (MMSE and IADL scores) and their neuropsychiatric symptom severity (NPI-Q score). Secondary outcomes include changes in caregiver HRQoL and psychological distress (PHQ-4).

Thus, this study expands upon existing evidence by prospectively monitoring a heterogeneous outpatient cohort—encompassing individuals with no cognitive impairment, MCI, and dementia—alongside their primary caregivers within a single rehabilitation setting over a one-year period. This design uniquely captures the longitudinal clinical transitions and their concurrent impact on the dyadic relationship. Furthermore, we adopt a multidimensional perspective, simultaneously evaluating cognitive, emotional, functional, and HRQoL domains in both patients and caregivers. By integrating these diverse variables, we aim to identify independent predictors of caregiver burden that extend beyond established demographic factors such as gender. Ultimately, this approach allows for a more granular exploration of how the interplay between clinical progression and psychosocial dynamics shapes caregiver outcomes over time—an area that remains insufficiently characterized in the current literature.

## 2. Materials and Methods

### 2.1. Study Design

This is a single-center, prospective longitudinal observational study conducted over a 4-year period (July 2019 to April 2023). The study monitors a cohort of outpatient–caregiver dyads attending the Cognitive Disorders and Dementia Center (CDCD) of a rehabilitation hospital. Participants were evaluated at baseline (T0) and re-evaluated at a 12-month follow-up (T1) to assess the progression of cognitive and functional impairment and its impact on caregiver burden and quality of life.

### 2.2. Participants

The present study took place in the Italian rehabilitation hospitalIstituti Clinici Scientifici Maugeri (Montescano)*,* specifically within the outpatients’ Care Pathway of Cognitive Disorders (Centro per i Disturbi Cognitivi e Demenze, CDCD). Based on inclusion and exclusion criteria, consecutive outpatients and caregivers were excluded if they presented one or more of the following conditions: visuo-perceptual deficits or significant speech impairments compromising the reliability of neuropsychological assessments; severe medical conditions (e.g., advanced respiratory or cardiac disease, neoplasms); a positive history of psychiatric disorders; non-Italian education or illiteracy; refusal to provide consent at any stage of the study.

A dedicated multidisciplinary team, including neurologists and neuropsychologists, screened consecutive outpatients attending CDCD, enrolling only those who met above mentioned eligibility criteria.

### 2.3. Procedure

Data regarding the baseline evaluation can be found in a previous paper [13]. The current article discusses data and results exclusively concerning the follow-up research and their comparison with baseline data. Patient recruitment, psychological and neuropsychological assessment, and data collection were performed by a multidisciplinary team of neurologists and psychologists (Table 1).

The study involved two phases:*(a)* *T_0_: Baseline Assessment*

The baseline evaluation, previously described [13] and reported in the methodological section below, was conducted during patients’ neurological examinations following their informed consent. Neurologists performed clinical assessments and diagnostic evaluation, while psychologists administered neuropsychological screenings, including the Mini-Mental State Examination (MMSE) [14,15,16] and the Addenbrooke’s Cognitive Examination-Revised (ACE-R) [17] if MMSE > 22, (ACE-R equivalent score ≤ 1 is considered potential MCI). Thus, patients were categorized as follows: No-CI (no cognitive impairment; n = 41), MCI (Mild Cognitive Impairment; n = 33), and DEM (dementia; n = 58). Additionally, caregivers aged > 65 years were screened with MMSE to confirm the absence of cognitive impairment.

All neuropsychological tests for patients and caregivers have already been described in detail in a previous paper [13] and briefly summarized in Table 1 (see also Appendix A). The neuropsychologists conducting the follow-up assessments were blinded to the patients’ baseline cognitive scores to minimize bias. The same standardized battery of tests was administered at both points in a quiet, dedicated clinical setting. Socio-demographic data, including age, education, occupation, and primary informal caregiver information (e.g., gender, civil status, informal caregiver’s reference person, such as wife/husband), were collected for all participants using a structured questionnaire. Of note, during the anamnestic collection, the patient and—in the case of an advanced patient’s deterioration—the caregiver were asked who the main and stable caregiver was. Thus, the primary caregiver was considered to be the one self-referred to by the dyad.

Patient frailty was assessed using Body Mass Index (BMI) and the Cumulative Illness Rating Scale (CIRS), focusing on the comorbidity index (CIRS-CI), which is the score defined as the number of categories in which a score of 3 or higher is recorded [18]. Thus, we evaluated the overall health status and comorbidities. These data were assessed only at T0 to characterize patients.

Psychological (e.g., depression, anxiety, and HRQoL) and functional assessments were conducted for MCI patients and caregivers of MCI/DEM patients. Caregivers also received psychoeducational sessions focused on illness management and were provided recommendations for tailored cognitive rehabilitation therapies for patients, both standard and technological [19]. In particular, in parallel with the assessment at T_0_ and T_1_, caregivers were offered a one-hour individual counseling session with a psychologist to discuss the specific challenges encountered during patient care. On these occasions, they received psycho-educational suggestions for managing the care burden and fostering the relationship with the patient. In addition, during the first meeting, caregivers received a printed booklet with practical tips for dealing with a person with cognitive impairment.

Participants without cognitive impairment (No-CI group) did not undergo further assessments but were monitored longitudinally (T_1_). Since the participants were independent in both basic and instrumental activities of daily living, no caregivers were included in the research.

*(b)* 
*T1: Telephone Interview and Follow-Up Assessment*


Twelve months after the baseline evaluation (T0), a telephone interview was structured to gather updated information on the patient’s clinical status and to verify whether caregivers were available for the in-person assessment. All patients initially enrolled in the study (n = 132) were considered to begin the follow-up phase. As outlined in Figure 1, 60 individuals were no longer eligible for follow-up (3 patients had died, while the remaining 57 could not be reassessed within the planned timeframe because the COVID-19 pandemic had shifted their follow-up dates beyond the acceptable window). These exclusions involved all three baseline cognitive groups (No CI = 20, MCI = 14, DEM = 23), thereby reducing the eligible sample to 72 individuals.

Among the 72 patients who completed the T1 telephone interview, 21 patients were excluded (7 died, 12 experienced a significative clinical deterioration, and 2 moved to a different clinical center).

In the end, 51 patients were able to attend the in-person follow-up assessment. Based on neuropsychological screening results (MMSE and ACE-R if MMSE > 22), patients and caregivers were reclassified into the same cognitive impairment group or reallocated to MCI or DEM categories if cognitive impairment had emerged or progressed since the baseline (Figure 1). Psychological and functional follow-up assessments were conducted and tailored to each group (according to Table 1).

**Table 1 brainsci-15-01316-t001:** Neuropsychological and functional assessment of respondents of three groups.

Instrument	Construct	Respondent	No CI	MCI	DEM
° Ad hoc schedule	Socio-demographic and clinical informations (age, education, occupation, primary caregiver; BMI, smoke, physical activity...)	Patient	√	√	√
		Caregiver		√	√
° CIRS	Comorbidity index	Patient	√	√	√
MMSE	Cognitive impairment screening tool	Patient	√	√	√
		Caregiver		√	√
* ACE-R	More in-depth screening tool for cognitive impairment	Patient	√	√	√
		Caregiver			
BADL	Autonomy levels in patient’s basic daily activities	Caregiver		√	√
IADL	Autonomy levels in patient’s instrumental daily activities	Caregiver		√	√
NPI-Q	Presence and intensity of patient’s neuropsychiatric symptoms	Caregiver		√	√
PHQ-4	Self-report screening scale for the assessment of distress	Patient		√	
		Caregiver		√	√
EQ-5D, EQ-VAS	Health-related quality of life	Patient		√	
		Caregiver		√	√
FSQ-SF	Caregiver burden	Caregiver		√	√

Note. Questionnaires’ references: Cumulative Illness Rating Scale (CIRS-CI) [18], Mini Mental State Examination (MMSE) [14]; Addenbrooke’s Cognitive Examination—Revised (ACE-R) [17]; Basic Activity of Daily Living (BADL) [20]; Instrumental Activity of Daily Living (IADL) [21]; Neuropsychiatric Inventory Questionnaire (NPI-Q) [22]; Patient Health Questionnaire–4 (PHQ-4) [23]; EuroQoL 5D (EQ-5D) and EuroQoL VAS (EQ-VAS) [24]; Family Strain Questionnaire—Short Form (FSQ-SF; cutoff > 6) [25]. * Only if MMSE > 22; ° Administered only at T0.

### 2.4. Sample Size Computation

The sample size calculation was based on the EQOL-VAS variable. We aimed for a sample large enough to detect a minimal difference of 15 points between caregivers of MCI and DEM patients, with a two-tailed Type I error of 5% and 80% power. An estimate of the standard deviation of 15 was used, based on data from the literature and preliminary data from our group. This corresponds to a Cohen’s d of 1.0, indicating a large effect size. Accordingly, the required sample size was determined to be 34 participants (17 + 17).

### 2.5. Statistical Analysis

The normality of the distribution of continuous variables was assessed using the Shapiro–Wilk test. Since several variables violated the normality assumption, central tendency and dispersion were reported as median (Q1, Q3), and hypothesis testing was performed using non-parametric procedures. Categorical data were presented as absolute and relative frequencies. Missing data were not replaced. Between-group comparisons of continuous variables were conducted using the Mann–Whitney U test or the Kruskal–Wallis test when comparing three groups. Categorical variables were compared using the Chi-square test or Fisher’s exact test, as appropriate. The association between caregiver HRQoL (EQ-VAS) and both caregiver distress and burden was assessed using Spearman’s correlation coefficient. To investigate the factors associated with caregiver burden, measured by the FSQ-SF, two separate stepwise multiple linear regression analyses were conducted at baseline (T0) and at follow-up (T1). A total of 15 candidate predictor variables were initially examined, including demographic, cognitive, functional, and clinical characteristics of both patients and caregivers. Patient variables included gender, age, education, MMSE score, BADL score, IADL score, comorbidity index, and total number of neuropsychiatric symptoms. Caregiver variables included gender, age, MMSE score, employment status, relationship to the patient, co-residence, and caregiver strain index. Model selection was based on changes in the Bayesian Information Criterion (BIC). The significance level for entering the model was set at 0.15, and the significance level for remaining in the model was 0.05. Multicollinearity was evaluated using the Variance Inflation Factor (VIF). Residual normality was assessed using the Shapiro–Wilk test, and homoscedasticity was tested using the White specification test, which evaluates whether residual variance remains constant across the range of fitted values and predictors. A two-tailed *p*-value < 0.05 was considered statistically significant. When appropriate, false discovery rate (FDR) was controlled at 5% using the Benjamini–Hochberg method, and FDR-adjusted *p*-values were also reported. All analyses were performed using SAS/STAT software, Version 9.4 (SAS Institute Inc., Cary, NC, USA).

## 3. Results

A total of 51 patients (mean age 80.0 ± 6.1 years) and 34 caregivers (mean age 58.8 ± 15.9 years) were evaluated at baseline (T0) and follow-up (T1). Notably, patient loss during the COVID-19 pandemic was observed (Figure 1), attributed to deaths from the disease and its complications, as well as the temporary closure of the CDCD outpatient clinic. Of these patients (n = 57), 49.12% (n = 28) remain under care at the Istituti Clinici Scientifici Maugeri CDCD, where they regularly undergo neurological examinations for cognitive disorders.

Table 2 and Table 3 summarize the anagraphic information (i.e., age, education), clinical, neuropsychological, functional, neuropsychiatric, psychological, and HRQoL variables for patients and caregivers at T0 and T1, along with comparisons where applicable.

### 3.1. Patient Characteristics

The study considered various types of dementia, with the following prevalence: vascular dementia (41.2%), Alzheimer’s disease (31.4%), frontotemporal dementia (5.9%), mixed dementia (9.7%), and Mild Cognitive Impairment (MCI; 11.8%). At baseline (T0), the ACE-R was administered to No-CI patients (n = 17), showing a median equivalent score of 3 (Q1–Q3: 2–4). At follow-up (T1), the same assessment was conducted in No-CI patients (n = 10; median equivalent score: 3 [Q1–Q3: 2–4]) and MCI patients (n = 13; median equivalent score: 0 [Q1–Q3: 0–1]).

MMSE, BADL, and NPI-Q severity scores were significantly worse in patients with dementia (DEM) compared to those with MCI at both T0 and T1. Conversely, IADL and NPI-Q symptom scores were significantly worse in DEM patients compared to MCI patients only at T0.

The patient cohort was evenly divided by gender (male: 52.9%, female: 47.1%), with the majority retired (96.1%). Most patients were married (62.7%) and relied on family members as primary caregivers, particularly their children (47.1%) or spouses (47.1%), with a small minority (5.8%) depending on other individuals (relatives, friends, or in-home nurses).

In terms of socio-economic characteristics, most patients possessed a disability exemption certification issued by the Italian Healthcare System, primarily due to low income and age ≥ 65 years (49.0%) or civil disability (15.7%). This certification has to be considered as an index of disability, reflecting disease severity and physical or economic frailty. Among these, dementia patients constituted the majority of those receiving disability benefits.

No significant differences were observed in BMI between No-CI, MCI, and DEM patients (Kruskal–Wallis χ^2^(2) = 2.34, *p* = 0.31). Similarly, comorbidity levels were not different among the three groups (Kruskal–Wallis χ^2^(2) = 0.927, *p* = 0.63), with high values (mean > 6, cut-off = 3) in both MCI and DEM groups.

### 3.2. Caregiver Characteristics

Table 3 reports the caregivers’ characteristics. The majority were female (85.4%), with nearly half still employed (46.4%) and the other half retired (49.8%), and most were married (68.3%). Many caregivers relied on their spouses (50.0%) for support. Caregiver education levels were slightly higher than those of patients (13 (8, 13) vs. 5 (5, 8) years, *p* = 0.002), and MMSE scores were generally above the normal cut-off at both time points.

At T1, caregivers of DEM patients reported significantly higher distress levels (PHQ-4) compared to caregivers of MCI patients. Furthermore, significant negative correlations were observed at both T0 and T1 between caregiver HRQoL (EQ-VAS) and both distress (PHQ-4; T0: r = −0.71, *p* < 0.0001, T1: r = −0.54, *p* = 0.0009) and caregiver burden (FSQ-F; T0: r = −0.67, *p* < 0.0001, T1: r = −0.44, *p* = 0.006) in both MCI and DEM groups.

### 3.3. Patient’s Cognitive Status Outcomes over Time

The comparison of cognitive status between T0 and T1 revealed statistically significant changes (Table 4). At T0, 17 patients were No-CI, but at T_1_, only 11 of them remained No-CI, while 4 progressed to MCI and 2 to DEM. From the MCI group at T_0_, 11 remained MCI at T_1_, while 4 worsened to DEM. All 19 DEM patients at T0 remained DEM at T1.

This clearly indicates a trend of deterioration over time, with patients tending to progress from milder conditions to more severe ones: No-CI to MCI or DEM and MCI to DEM.

Cognitive functioning showed both maintenance and decline over time: 17 No-CI patients at T0 progressed to MCI (n = 4) or DEM (n = 2) by T1, while 15 MCI patients at T0 evolved to DEM (n = 4) at T1. However, 11 MCI and 19 DEM patients remained stable.

### 3.4. Caregiver Outcomes over Time

For caregivers of MCI or dementia patients, there was no statistically significant change (T_0_: 6.68 ± 1.72 vs. T_1_: 6.42 ± 1.41; Delta = −0.26 ± 2.05; *p* = 0.49) in overall health-related quality of life between baseline and follow-up.

Moreover, caregiver’s burden level (FSQ-SF) remained constant over time as showed in Table 3, both in the MCI (T_0_: 13.2 ± 8.3 vs. T_1_: 11.7 ± 7.1, *p* = 0.62) and dementia groups (T_0_: 14.2 ± 5.5 vs. T_1_: 15.0 ± 7.2, *p* = 0.46), and in the total sample (T_0_: 13.8 ± 6.6 vs. T_1_: 13.8 ± 7.3, *p* = 0.88). Specifically, burden levels above the cutoff of 6 are considered at risk and require referral to a specialist psychologist or psychiatrist.

### 3.5. Predictors of Caregiver Burden

Potential predictors of caregiver burden at T0 and T1 were analyzed (Table 5).

At baseline (T0), the final model retained three significant independent predictors: the patient’s MMSE score, BADL score, and the number of neuropsychiatric symptoms (NPI-Q).

The overall model was statistically significant compared to a constant-only model (F(3, 29) = 6.07, *p* = 0.002) and accounted for approximately 32.2% of the variance in caregiver burden (Adjusted R^2^ = 0.322). Residuals were normally distributed (Shapiro–Wilk, *p* = 0.49), and homoscedasticity was confirmed (White test, *p* = 0.52).

At follow-up (T1), the regression model retained the same three patient-related predictors (MMSE, BADL, and NPI-Q symptoms) and additionally identified patient gender as a significant independent predictor. This indicates that, in addition to MMSE, BADL, and neuropsychiatric symptoms, being female was associated with higher caregiver burden at follow-up. The model was statistically significant (F(4, 28) = 4.26, *p* = 0.008), and explained approximately 29% of the variance in FSQ-SF scores (Adjusted R^2^ = 0.290).

Residuals were normally distributed (Shapiro–Wilk, *p* = 0.65), and homoscedasticity was confirmed (White test, *p* = 0.47). The Variance Inflation Factors (VIFs) were all <2.5, ruling out collinearity issues.

## 4. Discussion

This longitudinal study aimed to explore the impact of cognitive deterioration on the patient–caregiver dyad, with a focus on multidimensional factors influencing caregiver burden. Informal caregivers, despite being vital in managing cognitive disorders, are often unrecognized as part of the healthcare team. This oversight is significant because caregiving may negatively affect caregivers’ HRQoL, which, in turn, influences patients’ health outcomes [7,11,26]. The dyadic relationship in this context is multifaceted, involving caregivers’ efforts to maintain a sense of togetherness despite the challenges posed by cognitive deterioration. Its quality is largely shaped by the premorbid relationship and the caregiver’s mindset [27].

As reported in the literature and clinical practice, the patient cohort consisted predominantly of older adults with low education levels, significant comorbidities, frailty, and disability certifications, particularly among those diagnosed with dementia.

Moreover, consistently with previous research [26] and our first study [13], dementia patients showed worse clinical, neuropsychological, and functional outcomes compared to MCI patients, both at baseline (T0) and follow-up (T1). These differences were particularly evident in MMSE, BADL, IADL, and NPI-Q scores. Over the follow-up period, some patients maintained their cognitive status, while others progressed from no cognitive impairment to MCI or from MCI to dementia, in alignment with the literature [28]. At T_1_, dementia patients remained more impaired than MCI patients in MMSE, BADL, and the severity of behavioral symptoms [29]. Although these findings may be largely expected, the sustained and progressively increasing caregiving burden associated with the worsening of dementia remains a critical issue. This burden is not sporadic but persistent and can have a significant impact on the caregiver’s well-being, turning dementia into a family experience [11]. Therefore, it is essential to address this aspect through an early, multidisciplinary approach to ensure comprehensive support and care [12].

Focusing on the caregiver side, we highlighted some findings deserving of discussion. Caregivers were younger, mostly female, married, and active in the workforce, with slightly higher education levels and preserved cognitive function. They exhibited high levels of caregiving burden from the early stages of the care pathway, starting at the diagnosis of MCI or dementia. That is, caregiver burden was evident at diagnosis and remained consistently high over time, thus necessitating early caregivers’ psychological evaluation since the beginning of the dyad journey [11,30]. Indeed, this data highlights the chronic nature of the dyad involving a caregiver and a patient with cognitive decline. Moreover, caregivers experienced significantly reduced HRQoL, regardless of the patient’s cognitive impairment level. This aligns with previous findings showing that caregiving negatively correlates with HRQoL [31]. This may be primarily linked to the fact that a diagnosis of cognitive impairment necessitates a redefinition of family roles, requiring the son or daughter to assume the responsibility of caring for someone who once cared for them [26].

Additionally, the study shows that caregivers’ health-related quality of life, assessed at both baseline and follow-up, declines with increasing levels of distress, reinforcing previous research findings [5,7,12,32]. The quality of life in individuals with dementia and their caregivers has been shown to be interconnected and influenced by personal, partner, and dyadic factors. However, research on the mechanisms through which one member’s factors affect both their own and their partner’s quality of life within the dyad remains limited. This gap also applies to other psychosocial outcomes, such as depression and anxiety. Further investigation is needed to uncover the mechanisms by which dyadic members influence each other’s psychosocial well-being [12].

Regression analysis confirmed that patient factors such as MMSE, BADL, and NPI-Q symptoms were strong predictors of caregiver burden at both T_0_ and T_1_. Additionally, patient gender emerged as a significant factor at T_1_. Neuropsychiatric symptoms, including irritability, delusions, and agitation, were critical predictors of changes in caregiver burden over time, as were caregiver demographic and caregiving-related factors, such as co-residence, relationship status, and the number of caregiving hours [33,34].

### 4.1. Implications for Dyadic Interventions

Chronic illnesses such as dementia profoundly affect patients and caregivers, often leading to declines in physical health, mental well-being, and relationship quality [11,13]. From a preventive standpoint, identifying risk factors and triggers that affect caregivers of individuals with MCI and dementia is crucial [13]. This proactive approach can help reduce hospitalizations, delay institutionalization, and prevent the worsening of caregivers’ health, particularly in cases with pre-existing conditions. Moreover, dyadic interventions, including psychoeducational and health-focused programs, have been shown to improve HRQoL, emotional well-being, and caregiver self-efficacy at minimal financial cost [12,35,36]. Current data also reveal that the caregiver’s burden is significant from the outset, despite the severity of cognitive decline. In other words, the presence of cognitive decline is sufficient to put the caregiver’s health at risk. Therefore, the care of the patient–caregiver dyad should begin at diagnosis. Indeed, the dyadic perspective underscores the mutual influence between patients and caregivers. Despite its importance, caregivers’ experiences, costs, and outcomes are often overlooked in economic evaluations of dementia care [37]. Addressing this gap is essential for developing comprehensive care models that support both members of the dyad.

### 4.2. Impact of the COVID-19 Pandemic

The COVID-19 pandemic significantly affected the study by delaying follow-up assessments due to clinic closures and excluding some participants. Although its direct impact on cognitive impairment could not be confirmed, previous studies have shown that pandemic-related isolation and disruptions exacerbated cognitive decline in patients and heightened depressive symptoms in caregivers [38,39]. These findings align with the observation that neuropsychiatric symptom onset during the pandemic likely worsened caregiver burden [40].

### 4.3. Strengths and Limitations

This study’s strengths include its multidimensional assessment of patient–caregiver dyads, incorporating socio-demographic, clinical, and psychological data. It provides insights into how disease progression impacts both patient functioning and caregiver burden, and a continuum was identified between cognitive decline and changes in the patient–caregiver relationship. However, the study has several limitations. First, the small sample size and single-center design may restrict the generalizability of the findings beyond this specific rehabilitation setting. In addition, attrition related to the COVID-19 pandemic may have introduced attrition bias, and although participants were followed within a standardized time window, we did not perform a systematic comparison of baseline characteristics between those who completed the follow-up and those lost to follow-up. The lack of longitudinal screening of non-impaired patients and the absence of more detailed neuropsychological assessments to explore specific cognitive domains also limit the depth of our conclusions. Another limitation is that BMI and CIRS were assessed only at T0 due to clinical and time constraints; although relevant information was subsequently collected through self-reported data during telephone follow-ups and appeared stable, the absence of repeated objective measurements may have introduced additional bias.

Regarding the screening of caregivers over 65 years using the MMSE, this procedure may theoretically introduce selection bias by missing younger caregivers with cognitive impairment, although such cases are rare in our clinical practice. A further limitation concerns the No-CI group, which comprised cognitively healthy individuals who typically attended clinical visits alone; while their functional independence did not require caregiver assistance, including an informant assessment for this group would have strengthened the study design—particularly given the clinical relevance of subjective cognitive decline—and future research should incorporate collateral history even for non-impaired cohorts. Moreover, we did not quantify the time spent on caregiving activities across all groups, partly because of feasibility issues and the difficulty of obtaining precise estimates from caregivers, and we acknowledge that the counseling and psycho-educational interventions provided as part of routine care may have influenced caregiver outcomes, potentially introducing performance bias.

Future research should address these limitations and further investigate the mechanisms underlying dyadic interdependence in dementia care.

## 5. Conclusions

Despite the rising prevalence of cognitive impairment, the existing literature provides limited insights into the multifaceted factors influencing the psychosocial well-being of patient–caregiver dyads, as well as the potential actor–partner interdependence mechanisms.

This study adopts a multidimensional perspective on the dyadic relationship, emphasizing the importance of assessing psychosocial variables not only in patients with cognitive impairment but also in their caregivers. Our findings highlight the necessity of addressing both physical and psychological burdens to provide comprehensive support and patients’ cognitive training that fosters the psychosocial well-being of both members of the dyad [5].

It is, therefore, critical to investigate, assess, and monitor the HRQoL of both patients and caregivers over time. This approach aims to enhance the effectiveness of support services for individuals with cognitive impairment and assist caregivers in managing the physical and emotional challenges associated with caregiving [37].

Such efforts are particularly important given the adverse impact that MCI and dementia can have on patient–caregiver dyads.

Furthermore, healthcare policies should prioritize supporting caregivers through structured interventions, including psychoeducational programs that promote more functional and professional caregiving practices. These initiatives can improve caregiver resilience, reduce burden, and optimize outcomes for both patients and their families [35,36].

## Figures and Tables

**Figure 1 brainsci-15-01316-f001:**
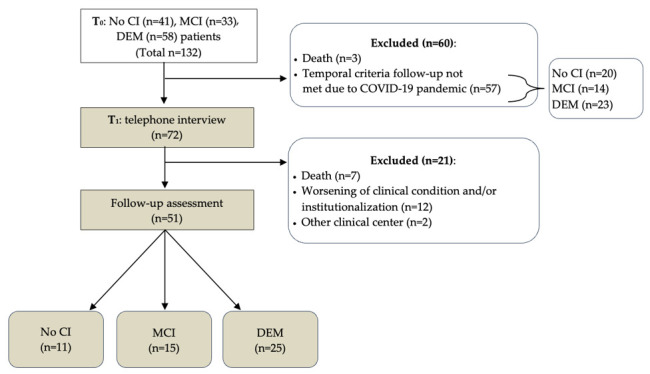
Flowchart of patient recruitment at baseline (T_0_) and follow-up (T_1_).

**Table 2 brainsci-15-01316-t002:** Comparison between No-CI subjects, MCI, and dementia patients’ group (T0 and T1).

**Variables (T0)**	**No-CI (n = 17)**	**MCI (n = 15)**	**DEM (n = 19)**	**MCI vs. DEM** ***p* Value**
Age	80.0 (75.8,83.3)	83.0 (78.0,83.0)	83.0 (76.0,85.0)	0.70
Female gender (%)	8 (47%)	5 (33%)	11 (58%)	0.34 *
Education	8.0 (5.0,8.0)	5.0 (5.0,11.7)	5.0 (5.0,8.0)	0.65
BMI	26.0 (24.8,28.3)	25.9 (23.4,27.7)	24.2 (21.6,27.1)	0.41
CIRS	5.5 (3.5,8.0)	4.0 (4.0,6.8)	5.0 (2.0,8.0)	0.86
MMSE	26.7 (24.7,27.6)	21.4 (20.5,23.5)	13.4 (10.9,17.4)	<0.0001 (<0.0001)
ACE-R * = 1	2 (16.7%)			
ACE-R * = 2	4 (33.3%)			
ACE-R * = 4	6 (50.0%)			
BADL °	6.0 (6.0,6.0)	6.0 (5.0,6.0)	5.0 (3.0,5.0)	0.017 (0.05)
IADL °	4.0 (4.0,4.0)	4.0 (2.0,5.0)	1.0 (0.0,2.0)	0.004 (0.007)
NPI-Q—Symptoms °	3.0 (3.0,3.0)	3.0 (1.0,5.0)	4.0 (3.0,5.8)	0.048 (0.08)
NPI-Q—Severity °	3.0 (3.0,3.0)	4.0 (1.0,9.0)	8.0 (5.0,11.5)	0.028 (0.06)
NPI-Q Discomfort °	2.0 (2.0,2.0)	3.5 (0.0,14.0)	7.0 (3.8,9.8)	0.19
PHQ-4		3.5 (1.0,5.0)		
EQ-5D-3L		7.0 (5.3,8.0)		
EQ-VAS		60.0 (42.5,70.0)		
**Variables** **(T1)**	**No-CI (n = 11)**	**MCI (n = 15)**	**DEM (n = 25)**	**MCI vs. DEM** ***p* Value**
Age	80.0 (76.5,82.5)	82.0 (78.0,83.8)	83.0 (75.8,85.0)	0.75
Female gender (%)	7 (64%)	3 (80%)	14 (56%)	0.039 † (0.062)
MMSE	26.7 (23.8,27.4)	21.4 (20.6,22.6)	9.6 (4.3,13.7)	0.0002 (0.0014)
ACE-R * = 0		9 (69.2%)		
ACE-R * = 1		4 (30.8%)		
ACE-R * = 2	5 (50.0%)			
ACE-R * = 3	2 (20.0%)			
ACE-R * = 4	3 (30.0%)			
BADL °		6.0 (5.0,6.0)	3.0 (1.0,6.0)	0.004 (0.016)
IADL °		3.0 (1.0,5.0)	1.0 (0.0,3.0)	0.025 (0.052)
NPI-Q—Symptoms °		3.0 (0.3,4.8)	4.0 (2.0,6.0)	0.14
NPI-Q—Severity °		4.0 (0.3,6.0)	6.0 (4.5,11.0)	0.028 (0.056)
NPI-Q Discomfort °		2.0 (0.3,8.5)	7.0 (3.0,11.3)	0.13
PHQ-4		3.0 (2.0,5.2)		
EQ-5D-3L		7.5 (6.0,10.0)		
EQ-VAS		67.5 (45.0,83.0)		

Note: ° caregiver’s answers, * administered only in patients with MMSE > 22. Data are reported as median (Q1, Q3) or as n (%). Reported *p*-values are from the Mann–Whitney U test, except †: Fisher’s exact test. FDR-adjusted *p*-values are reported in brackets.

**Table 3 brainsci-15-01316-t003:** Comparison between caregivers of MCI and dementia group (T_0_ and T_1_).

**Variables (T_0_)**	**Total (n = 34)**	**MCI (n = 15)**	**DEM (n = 19)**	* **p** *
Age	56.0 (49.0,75.0)	56.0 (43.3,71.8)	57.0 (53.3,77.3)	0.26
Female Gender (%)	29 (85%)	13 (87%)	16 (84%)	0.99 †
Education	13.0 (5.0,13.0)	8.0 (5.0,13.0)	13.0 (8.0,13.0)	0.44
MMSE	29.5 (27.3,30.0)	29.0 (27.1,30.0)	30.0 (27.3,30.0)	0.47
PHQ-4	4.0 (1.0,6.0)	3.5 (1.0,5.0)	4.0 (1.0,6.0)	0.61
EQ-5D-3L	6.0 (5.0,8.0)	6.0 (5.0,7.0)	6.0 (6.0,8.0)	0.34
EQ-VAS	80.0 (65.0,90.0)	82.5 (60.0,95.0)	75.0 (65.0,85.0)	0.39
FSQ-SF	14.0 (9.8,19.0)	14.0 (4.0,20.0)	13.0 (10.3,17.8)	0.99
**Variables (T_1_)**	**Total (n = 40)**	**MCI (n = 15)**	**DEM (n = 25)**	* **p** *
Age	56.5 (49.0,74.0)	52.0 (41.0,75.0)	60.0 (54.0,71.3)	0.11
Female Gender (%)	32 (80%)	10 (67%)	22 (88%)	0.63 †
Education	13.0 (7.2,13.0)	13.0 (5.0,13.0)	10.5 (8.0,13.0)	0.86
MMSE *	28.5 (26.6,30.0)	28.7 (27.0,30.0)	28.5 (26.1,30.0)	0.67
PHQ-4	4.0 (1.0,5.0)	1.5 (1.0,4.0)	4.0 (3.0,5.8)	0.026 (0.21)
EQ-5D-3L	6.0 (5.0,7.0)	6.0 (5.0,6.0)	6.0 (5.0,7.0)	0.16
EQ-VAS	80.0 (60.0,92.8)	80.0 (71.3,89.8)	72.5 (60.0,95.0)	0.52
FSQ-SF	15.5 (7.0,19.0)	9.0 (7.0,18.5)	18.0 (8.5,20.0)	0.22

* Only caregivers over 65 years old (n = 12). Data are reported as median (Q1, Q3) or as N (%). Reported *p*-values are from the Mann–Whitney U test, except †: Chi-square test. FDR-adjusted *p*-values are reported in brackets.

**Table 4 brainsci-15-01316-t004:** Patient sample comparison between baseline and follow-up assessments.

Variables	No-CI T1	MCI T1	DEM T1		Total
No CI T0	11 (64.7%)	4 (23.5%)	2 (11.8%)	17	(100%)
MCI T0	0	11 (73.3%)	4 (26.7%)	15	(100%)
DEM T0	0	0	19 (100.0%)	19	(100%)
Total	11 (21.6%)	15 (29.4%)	25 (49.0%)	51	(100%)

Note: Transition matrix of cognitive status from baseline (T0) to follow-up (T1). Rows represent cognitive status at T0, and columns represent status at T1. Percentages are calculated within rows. A significant association was found between baseline and follow-up cognitive status (Fisher’s Exact test, *p* < 0.0001).

**Table 5 brainsci-15-01316-t005:** Stepwise regression results for FSQ-SF caregiver burden at baseline (T_0_) and follow-up (T_1_).

**Variables T0**	**Estimate**	**95% CI**	**SE**	**t-** **Stat**	***p* Value**
MMSE	0.46945	0.09580, 0.84310	0.18269	2.570	0.016
BADL	−2.4678	−4.17575,−0.75994	0.83507	−2.955	0.006
NPI-Q Symptoms	1.1762	0.19919, 2.15320	0.4777	2.462	0.020
**Variables T1**					
Gender (Females)	5.083	0.62218,9.54392	2.1777	2.334	0.027
MMSE	0.52037	0.10641, 0.93434	0.20209	2.575	0.016
BADL	−1.851	−3.70717, 0.00524	0.90617	−2.043	0.051
NPI-Q Symptoms	1.2593	0.18929, 2.32938	0.52238	2.411	0.023

## Data Availability

The datasets generated and analyzed during the current study are available from the corresponding author upon reasonable request due to patient privacy considerations.

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
