# Peer review of "A Longitudinal Observational Study to Monitor the Outpatient–Caregiver Dyad in a Rehabilitation Hospital: Sociodemographic Characteristics and the Impact of Cognitive and Functional Impairment"

_brainsci, 2025, doi:10.3390/brainsci15121316_

Round 1
Reviewer 1 Report
Comments and Suggestions for Authors
This observational study aims to monitor the evolution of the patient–caregiver dyad in a rehabilitation hospital. Patients were classified at baseline as cognitively normal (No‑CI), mild cognitive impairment (MCI) or dementia (DEM) based on the Mini‑Mental State Examination (MMSE) and the Addenbrooke’s Cognitive Examination – Revised (ACE‑R). Caregivers were required to have preserved cognition. Authors conclude that caregivers experience significant burden and reduced health‑related quality of life (HRQoL) irrespective of the severity of cognitive impairment and that patient factors (MMSE, basic activities of daily living – BADL – score, neuropsychiatric symptoms) and patient gender predict caregiver burden.
Below I highlighted major issues mainly based on the STROBE checklist:
- The title identifies the study as monitoring a dyad but does not indicate that it is a longitudinal observational study. The abstract should state the study design and sampling frame. At present the abstract lacks essential details such as dates of recruitment, number of eligible participants, number lost to follow‑up and adjustments made for confounders.
- The introduction provides an overview of cognitive decline and caregiver burden but does not clearly articulate the knowledge gap addressed by this study. Prior literature already shows that most dementia caregivers are women and that female caregivers report higher burden and depression (https://pmc.ncbi.nlm.nih.gov/articles/PMC11334129/#:~:text=literature%20on%20the%20burden%20of,1). The authors should clarify how their study adds to existing evidence.
- The objective is to investigate socio‑demographic, clinical and psychological factors influencing caregiver burden. However, the authors do not state an a priori hypothesis or specify primary/secondary outcomes. Clear objectives are essential for selecting appropriate statistical methods and avoiding data‑driven conclusions.
- Although described in the methods section, the design is not succinctly stated early in the paper. It appears to be a single‑centre prospective cohort study. A clearer description of the design, duration and setting should be presented in the early methods section.
- The authors state that recruitment occurred at a single Italian rehabilitation hospital (ICS Maugeri) between July 2019 and April 2023 but do not describe the catchment population or how patients were referred. It is unclear whether this is a convenience sample, which limits generalizability.
- Eligibility criteria exclude patients with visuo‑perceptual deficits, speech impairments, severe medical conditions or psychiatric disorders. However, the process of screening and enrolment is not described. Figure 1 summarises attrition, yet the total number approached and reasons for non‑participation are missing. The requirement that caregivers over 65 years undergo MMSE may introduce selection bias because younger caregivers with cognitive impairment might be missed. In addition, the authors did not justify the inclusion of only patients with an informal caregiver (the No‑CI group had no caregiver data).
- The exposures and outcomes are inadequately defined. For example, caregiver burden is measured using the Family Strain Questionnaire (FSQ‑SF), but the psychometric properties or cut‑offs are only briefly mentioned. The authors should specify which variables are considered exposures (e.g., patient cognitive status, functional scores) and which are outcomes (caregiver burden, HRQoL), and provide definitions and validity of instruments.
- The manuscript refers readers to a previous publication for details of neuropsychological tests. The timing and standardization of assessments at baseline and follow‑up should be explained, including whether interviewers were blinded to patient status.
- Potential sources of bias are not addressed. This single‑centre cohort may not represent the broader population. Attrition bias is likely because of the COVID‑19 pandemic; yet there is no comparison of baseline characteristics between completers and those lost to follow‑up. The authors provided counselling and psycho‑educational interventions during the study, which may influence caregiver outcomes (performance bias); this needs to be acknowledged.
- No sample size calculation is provided. The final sample included 51 patients and 34 caregivers, which is small and may be underpowered for multivariable analyses. The authors should report the expected effect sizes and justify the adequacy of their sample.
- The statistical analyses raise several issues:
- Continuous variables violating normality were still summarized as mean ± SD, and parametric tests were used. Given the small sample, non‑parametric methods would be more appropriate or at least results should be corroborated by non‑parametric tests.
- The regression analysis uses a stepwise selection based on the Bayesian information criterion. Stepwise methods are discouraged because they inflate type I error and produce unstable models. Important confounders (e.g., caregiver age, relationship, employment status, co‑residence) were excluded yet may influence burden.
- The authors treated multiple comparisons (e.g., numerous t‑tests across tables) without adjustment for multiplicity, increasing the risk of false positives.
- There is no description of handling missing data. Given the number of variables measured and attrition, this could bias estimates.
- The results section includes tables comparing groups at baseline and follow‑up and describes regression outcomes. However: the text does not report numbers at each stage (eligible, included, followed, analysed) as recommended. Table 2 and Table 3 include numerous variables but the narrative summarises only a few. An overabundance of statistical tests may distract from key findings. The regression results show that patient MMSE and neuropsychiatric symptoms are associated with caregiver burden, but confidence intervals are not reported, making it difficult to judge precision. It is also unclear if model assumptions (linearity, homoscedasticity) were checked. There is no stratification by sex/gender of caregivers, despite evidence that female caregivers experience higher burden
Reviewer 2 Report
Comments and Suggestions for Authors
This study offers important insights into the dynamic relationship between cognitive decline in patients with MCI and dementia and the consequent burden experienced by their caregivers. By utilizing a comprehensive, longitudinal approach, the authors examine how patient socio-demographic, clinical, and psychological factors affect caregiver burden and health-related quality of life (HRQoL). The study's robust methodology, involving neuropsychological assessments at baseline and follow-up, as well as a thorough evaluation of both patient and caregiver characteristics, offers valuable information regarding the progression of cognitive impairment and its impact over time.
However, while the study provides meaningful results, some aspects require further clarification. The sample size remains a significant limitation, especially considering the high attrition rate due to the COVID-19 pandemic, which impacted follow-up assessments. While the data offer important insights into the predictors of caregiver burden, the small number of participants, particularly in the caregiver subgroups, may limit the generalizability of the findings. Furthermore, although the study effectively identifies key predictors such as cognitive impairment, functional abilities, and neuropsychiatric symptoms, the analysis could benefit from more nuanced exploration of the mechanisms underlying these relationships.
One of the strengths of the study lies in its longitudinal design, which allows the authors to track changes in caregiver burden and HRQoL over time. The findings that patient cognitive status and neuropsychiatric symptoms were strong predictors of caregiver distress and burden are consistent with existing literature, reinforcing the importance of addressing both patient and caregiver needs in intervention planning. However, the study would benefit from a more detailed analysis of the psychosocial factors influencing caregiver experiences and the long-term impact of interventions designed to support caregivers.
The study underscores the necessity of early and continuous support for caregivers of individuals with cognitive impairment. The authors suggest that psychosocial support, tailored to the needs of both patients and caregivers, is crucial for improving outcomes. This is particularly relevant in light of the significant findings regarding the emotional and physical strain experienced by caregivers, irrespective of the severity of the patient's cognitive impairment. As the study highlights, addressing caregiver well-being is essential for optimizing the care and quality of life for both patients and their caregivers. Future research should consider larger sample sizes and explore further the interplay between clinical, psychological, and socio-demographic factors in influencing caregiver burden.
Reviewer 3 Report
Comments and Suggestions for Authors
Thank you for the opportunity to review this insightful manuscript. The study addresses a critical area—the factors influencing caregiver burden and quality of life across the MCI/dementia spectrum over time—and the findings promise to be valuable.
The following major and minor points should be considered to strengthen the paper’s methodological completeness and clarity for publication.
1. Introduction section
Minor Point: Clarifying Care Models (Lines 68–71)
The discussion highlights the "shared care dyad" and "collaborative partnership." This section should be slightly expanded to ground the discussion in established care philosophies. Specifically, clarifying which model best aligns with the described partnership—such as person-centered care, patient-centered care, or family-centered care—would strengthen this study’s conceptual foundation.
2. Hypothesis and Statistical Methods
The core concern lies in the mixed use of parametric and non-parametric statistics, especially considering the sample size constraints. The statistical approach needs to be rigorously justified to ensure the reliability of the presented results.
A. Robustness of Group Comparisons (Objectives 1 & 2)
i. Selection of Test Methods for Continuous Variables
Query: The manuscript states that while Shapiro–Wilk tests indicated violations of the normality assumption, the parametric tests (t-tests, ANOVA) were used as the primary analysis because the violations were deemed "not severe."
Recommendation: Given the small sample sizes in the comparison groups (Group A, n = 11; Group B, n = 15; Group C, n = 25), parametric tests may be less robust to deviations from normality. To ensure statistical integrity, I recommend that for all variables where normality was violated, the non-parametric test results (Mann–Whitney U test, Kruskal–Wallis analysis) be presented as the primary findings. Additionally, due to the small sample size, for continuous variables, please use the median and IQR (interquartile range).
ii. Constraint on Categorical Data Analysis (Chi-square test)
Issue: The chi-square test was used to compare categorical data. However, cells with expected values of less than 5 constitute over 20% of all cells. This level of violation compromises the reliability of the Chi-square test's p-values.
Recommendation: Please reanalyze all affected categorical comparisons using a more appropriate method, such as Fisher’s exact test or the likelihood ratio chi-square test, and update the results in the tables and text accordingly.
B. Multiple Linear Regression Analysis (Objective 3)
i. Sample Size and Model Stability
Query: The stepwise multiple linear regression analysis for caregiver burden utilized 13 candidate predictor variables. Given this large number of initial candidates, please clarify the following:
- The total sample size (N) included in the multiple linear regression analysis is specified.
- Provide statistical justification that the ratio of the final number of independent variables to N is adequate to ensure the resulting model’s stability, statistical power, and generalizability.
ii. Residual analysis and assumptions
Recommendation: Please add verification results concerning the core assumptions to confirm the linear regression model’s robustness:
- Normality of residuals (e.g., results of the Shapiro–Wilk test on the residuals, or visual confirmation via QQ plots).
- Homogeneity of variance (Homoscedasticity) (e.g., confirmation from residual plots)
If these assumptions are violated, robust alternatives, such as robust regression or bootstrapping, are used to validate the parameter estimates and their standard errors.
3. Results Presentation (Tables)
A. Clarity and Structure in Tables 2 & 3
Recommendation: To clearly and efficiently communicate Objective 2 results (comparison between baseline and follow-up):
- Please restructure Tables 2 and 3 such that the baseline (T0, left side) and follow-up (T1, right side) data are placed adjacent to one another. This arrangement will make the change in values and the comparison over time much clearer to the reader.
- Consistent with the abovementioned methodological recommendation, please ensure the non-parametric test values (p-values) should be prominently displayed in these tables.
B. Table 3 Redundancy
The "Total (n=34)" column in Table 3 appears redundant because the sample size is the sum of the MCI (n = 15) and DEM (n = 19) groups. I recommend removing the "Total" column to streamline the presentation and focus on the key group comparisons.
C. Table 4 Caption and statistical integrity
Table 4 note section reports a significant association using the chi-square test (χ² = 54.0, p<0.0001). This result is statistically questionable because of the violation of the expected cell count.
Recommendation: This result must be re-evaluated using a robust test (Fisher’s exact or likelihood ratio), and the caption, table, and related text should be updated to reflect the new, reliable statistical outcome.
Round 2
Reviewer 1 Report
Comments and Suggestions for Authors
Thanks to the authors for providing the detailed explanations/editions of the raised issues.
Author Response
We sincerely thank the reviewer for the time and effort devoted to the evaluation of our manuscript and for the positive appreciation of our revisions.
Reviewer 2 Report
Comments and Suggestions for Authors
Dear authors, thank you for considering my comments.
Author Response

(The authors gave the same response as above.)

Reviewer 3 Report
Comments and Suggestions for Authors
Thank you for revising the manuscript according to the reviewers' comments. I have a few minor suggestions for further improvement below.
1. Introduction section:
Please review the usage of the abbreviation (HRQoL). I suggest you improve the sentence structure, line breaks, and conjunctions. The current phrasing feels somewhat redundant and could be improved for clarity and conciseness.
2. Tables:
Table 2 Note: Please correct the double periods (..) found in the Note.
Table 3 Note: Please ensure the opening word of the sentence is capitalized, as per standard formatting.
3. Statistical Notation (Use of Commas):
Additionally, please consider adding a comma when reporting statistical results in the main text for improved clarity and consistency.
For example, on Line 404, please consider revising:
(Fisher's Exact test p < 0.0001)
to:
(Fisher's Exact test, p < 0.0001).
Similarly, on Lines 415-416, please change:
Residuals were normally distributed (Shapiro-Wilk p = 0.49), and homoscedasticity was confirmed (White test p = 0.52).
to:
Residuals were normally distributed (Shapiro-Wilk test, p = 0.49), and homoscedasticity was confirmed (White test, p = 0.52).
Please review all other statistical notations in the manuscript for similar consistency.
I hope these comments are helpful.
